# Stem Cell-Based Therapy: A Promising Treatment for Diabetic Foot Ulcer

**DOI:** 10.3390/biomedicines10071507

**Published:** 2022-06-25

**Authors:** Racha El Hage, Uwe Knippschild, Tobias Arnold, Irene Hinterseher

**Affiliations:** 1Department of Vascular Surgery, Universitätsklinikum Ruppin-Brandenburg, Medizinische Hochschule Branderburg Theodor Fontane, Fehrbelliner Str. 38, 16816 Neuruppin, Germany; racha.elhage@mhb-fontane.de; 2Department of General and Visceral Surgery, Surgery Center, Ulm University, Albert-Einstein-Allee 23, 89081 Ulm, Germany; uwe.knippschild@uniklinik-ulm.de (U.K.); tobias-2.arnold@uni-ulm.de (T.A.); 3Berlin Institute of Health, Vascular Surgery Clinic, Charité-Universitätsmedizin Berlin, Freie Universität Berlin and Humboldt-Universität zu Berlin, 10117 Berlin, Germany; 4Fakultät für Gesundheitswissenschaften Brandenburg, Gemeinsame Fakultät der Universität Potsdam, der Medizinischen Hochschule Brandenburg Theodor Fontane und der Brandenburgischen Technischen Universität Cottbus—Senftenberg, Karl-Liebknecht-Str. 24-25, 14476 Potsdam, Germany

**Keywords:** diabetic foot ulcer (DFU), stem cell therapy, peripheral artery disease (PAD), secretome, nanomaterials

## Abstract

Diabetic foot ulcer (DFU) is a severe complication of diabetes and a challenging medical condition. Conventional treatments for DFU have not been effective enough to reduce the amputation rates, which urges the need for additional treatment. Stem cell-based therapy for DFU has been investigated over the past years. Its therapeutic effect is through promoting angiogenesis, secreting paracrine factors, stimulating vascular differentiation, suppressing inflammation, improving collagen deposition, and immunomodulation. It is controversial which type and origin of stem cells, and which administration route would be the most optimal for therapy. We reviewed the different types and origins of stem cells and routes of administration used for the treatment of DFU in clinical and preclinical studies. Diabetes leads to the impairment of the stem cells in the diseased patients, which makes it less ideal to use autologous stem cells, and requires looking for a matching donor. Moreover, angioplasty could be complementary to stem cell therapy, and scaffolds have a positive impact on the healing process of DFU by stem cell-based therapy. In short, stem cell-based therapy is promising in the field of regenerative medicine, but more studies are still needed to determine the ideal type of stem cells required in therapy, their safety, proper dosing, and optimal administration route.

## 1. Introduction

Diabetes mellitus is a major public health problem that represents one of the greatest medical emergencies worldwide. Diabetes is characterized by elevated levels of blood glucose, which leads, over time, to serious damage to the heart, blood vessels, eyes, kidneys, and nerves [1]. Vascular complications are considered the most serious indicators of diabetes mellitus, thereby representing the leading cause of morbidity and mortality in diabetic patients [2]. Other mediators of diabetes that cause vascular complications include dyslipidemia, chronic hyperglycemia, and insulin resistance. Dyslipidemia and chronic inflammation are the major causes for the development of atherosclerosis, which cause chronic accumulation of lipid-rich plaque in the arteries in diabetic patients [3]. Atherosclerosis is the main reason for reduced life expectancy in patients with diabetes, while diabetic retinopathy and nephropathy are the largest contributors to blindness and end-stage renal disease, respectively [4]. Prevention methods for vascular complications in diabetic patients include intensive blood glucose reduction, which in turn leads to lowering the risk of nephropathy and retinopathy, in addition to antihypertensive medicine and statin therapy that decrease the risk of cardiovascular disease. However, insulin resistance and its biological effects in various tissues may be more important factors than hyperglycemia in mediating atherothrombotic complications, particularly in type 2 diabetes [4]. Therefore, diabetes mellitus is not only considered a disorder of carbohydrate metabolism, but also the reason for vascular disease affecting mostly all types and sizes of blood vessels [5]. Another consequence of diabetes mellitus is microangiopathy that occurs in the colon, which has been reported to be more common in diabetics than non-diabetics [6]. Moreover, foot infections are other frequent complications of diabetes mellitus that are associated with high morbidity, occasional mortality, and heavy resource utilization, including antibiotic therapy and surgical procedures [7]. Ulcers in diabetics are usually infected with a variety of Gram-positive microorganisms such as *Staphylococcus aureus*, *Enterococcus*, and Gram-negative organisms such as *Pseudomonas aeruginosa*, *Escherichia coli*, *Klebsiella species*, *Proteus species*, etc., and other anaerobes [8]. These microorganisms can combine together and form biofilms which are the main cause of diabetic foot ulcer [8]. Diabetic foot is considered one of the most major and distressing complications of diabetes, and it is defined as a foot with ulceration that is associated with neuropathy and/or peripheral artery disease (PAD) of the lower limb in a diabetic patient [8]. This condition is more common in elderly patients [9]. It has been reported that the yearly incidence of diabetic foot ulcers (DFUs) is about 2% with a lifetime incidence between 19% and 34%, and about half of these ulcers become infected [10]. Approximately 26 million people worldwide annually have a DFU with another 130 million at risk of diabetic peripheral neuropathy [10]. Around 20% of moderate and severe diabetic foot ulcers result in amputation, making DFU the leading global cause of amputation, hospitalization, and disability, which in turn affects life quality [7]. Due to its multifactorial etiology, the treatment of diabetic foot ulcer is challenging. It is also considered to be a burden on patients, hospitals, healthcare systems, and society due to its high risk of recurrence [10]. Even when an ulcer has healed successfully, reported recurrence rates are 40% in the first year, and 65% in the first 3 years post healing [11]. Standard treatment methods for diabetic foot ulcer include optimization of blood flow, debridement, infection control, and offloading [12]. With these treatments, only 50% of patients heal within 20 weeks and 50% recur within 18 months, which urges the need for improved therapies [12].

## 2. Diabetes Mellitus and Its Associated Diseases

The major diseases associated with diabetes mellitus are diabetic polyneuropathy, peripheral vascular disease, macroangiopathy, and microangiopathy.

### 2.1. Peripheral Polyneuropathy

Diabetic neuropathies are a varied group of pathological indicators that can affect every organ. They can cause clinical complications such as organ dysfunction leading to low quality of life and increased morbidity [13,14]. Diabetic polyneuropathy (DPN) is described as the dysfunction of multiple peripheral sensory and motor nerves that branch out from the spinal cord into the hands, arms, legs, and feet. The longest nerves that extend from the spine to the feet are usually the most affected [15]. DPN is described by some authors as the most frequent microvessel diabetic complication, and it has been reported to be present in approximately 10% of diagnosed diabetic patients [14,16,17]. In addition, diabetic polyneuropathy is characterized by three main pathological features including inflammation, oxidative stress, and mitochondrial dysfunction that in turn affect microvessels and nerve fibers [14]. Effects of DPN include: unusual sensations (paresthesias) such as tingling, burning, or prickling; numbness and pain in hands, legs, and feet; weakness of muscles in feet and hands; sharp pains or cramps; extreme sensitivity to touch; insensitivity to pain or temperature changes; loss of balance or coordination, and difficulty in walking on uneven surfaces [15]. Due to the fact that DPN inhibits the ability to sense problems, it can put a person at risk for injuries in the feet and toes. Furthermore, it can also lead to ulcer development, wounds, and chronic infection in the feet. If left untreated, DPN can cause nerve damage to other parts of the body like the eyes, digestive tract, and sexual organs. DPN is also considered a primary reason for amputations [15].

### 2.2. Peripheral Vascular Disease

Peripheral vascular disease (PVD) is a slow and progressive blood circulation disorder that causes blood vessels around the heart and the brain to narrow blockages or spasms [18]. This disorder can happen in arteries, veins, or lymphatic vessels. Organs supplied by these vessels like legs and feet may not receive proper blood flow, which makes them the most affected. The most common cause of PVD is atherosclerosis, which is caused by the build-up of plaque inside the arterial wall. Plaques limit the blood and oxygen flow to the organs and limbs [19]. As plaques progress, blood clots may further decrease the inner size of the blood vessel, finally leading to complete blockage of major arteries. If left untreated, this can result in organ damage and loss of limbs, fingers, or toes [19]. PVD is usually the most common complication in both type 1 and type 2 diabetes [20]. Peripheral arterial disease (PAD) is a form of PVD that develops only in arteries carrying oxygen-rich blood away from the heart [19]. PAD is defined as stenosis or blockage of the lower limb arteries. PAD is considered one of the major public health problems resulting from macrovascular complications in diabetic patients; it occurs in half of all patients with diabetic foot ulcer and often leads to amputations. The risk of amputation for patients with diabetes is estimated to be 10–15%, which is 10–30 times higher than in the general population. The German Association of Angiology has reported around 60,000 cases of amputations annually, making Germany the highest with amputation incidence among other European countries. DFU and atherosclerosis are considered the main causes of amputation [21]. The main cause of PAD is atherosclerosis, which increases notably in diabetics [22]. The Centers for Disease Control and Prevention (CDC) has reported that approximately 12 to 20% of people over the age of 60 develop PAD [19]. Once PAD occurs in diabetic patients, the prognosis is poor. It has been reported that over a 5-year period, approximately 20% of affected patients have a non-fatal cardiovascular event, 4% undergo amputation, and 30% die [20,23]. The prevalence of PAD in diabetics is twice that as in non-diabetics. Accordingly, it is recommended by the American Diabetes Association (ADA) that the ankle-brachial index (ABI) should be performed as a measure of detection in patients older than 50 years of age who have suffered from the disease for more than 10 years [22,24]. PAD is a major predictor of foot ulcer in diabetic patients. Therefore, it has been recommended that the physician examining patients with diabetic foot ulcer should always assess the vascular status of lower limbs and search for signs of ischemia since around 50% of these patients have PAD [22]. However, due to the altered presentations of PAD and the limited diagnostic procedures, PAD detection, and severity assessment in diabetic patients with foot ulcer is a clinical challenge. In addition, wound healing in these patients is not only affected by PAD, but also by different factors such as infection and the existence of other comorbidities [22,25,26]. Furthermore, patients affected by DFU are often late to seek specialized medical care and are not treated in vascular centers. These factors, along with the mentioned pathologies, limit the possibilities for revascularization. Overall, diabetic patients have poor survival and are fivefold more likely to develop critical limb ischemia than non-diabetic patients [21,27,28].

### 2.3. Macroangiopathy

Diabetic macroangiopathy is a specific form of accelerated atherosclerosis. It is characterized by intraplaque new vessel formation due to abnormal angiogenesis, increased vascular permeability of capillary vessels, and tissue edema, frequently resulting in atherosclerotic plaque hemorrhage and rupture in addition to cardiac microvascular dysfunction [29,30]. The prematurity and rapidly progressive nature of atherosclerosis in diabetes are related to different mechanisms that are still unclear. Nevertheless, mechanisms related to hyperglycemia are playing a major role as they are linked to dysregulated vascular regeneration. Other key events involved are the activation of inflammatory pathways triggered by a dysregulated activation of membrane channel proteins, aquaporins, in addition to oxidative and hyperosmolar stresses [30]. Vascular dysfunction in diabetes is linked to different pathogenic mechanisms that include dysregulated regeneration of vessels, or impaired function of cells that are involved in the preservation of vascular homeostasis and permeability. These cells include smooth muscle cells, stromal cells, endothelial cells, pericytes, inflammatory cells, and circulating and tissue-resident vascular stem/progenitor cells [31]. Stem cells residing in the vessel wall or circulating in the blood have been considered lately as a mechanism for vascular damage repair and for the replacement of exfoliated endothelial cells. Disruption in the number of stem/progenitor cells circulating has been linked to cardiovascular risk factors and rapid progression of vascular disease [32,33]. Diabetes and its associated factors such as hyperglycemia and insulin resistance have been linked to the reduction or loss of function of progenitor cells in addition to stem cell mobilization defect, which is known as bone marrow mobilopathy [34]. Bone marrow mobilopathy occurs due to the failure of departure of stem/progenitor cells from the bone marrow into the peripheral circulation, which contributes to vascular disease [30,32,33]. Mesenchymal stem cells (MSCs) have been suggested as a possible target for diabetes-related pathogenic mechanisms [35]. The reduction in the abundance and function of stem cells circulating and in tissues including MSCs has been presented in both type 1 and type 2 diabetes diseases [32,33]. As a result, macrovascular complications in diabetes may reflect stem cell vasculopathy. Stem cell vasculopathy is described as the inability of the defective stem cell compartment to regenerate dying endothelial or vascular smooth muscle cells, or where this defective stem cell compartment contributes to macrovascular complications. It has been reported that in type 2 diabetes, the reduction in vascular stem cells is related to the degree of glycemic control [36,37], and some reduction of these cells has been linked with impaired glucose regulation prediabetes [30,38,39]. Preclinical models in diabetic macroangiopathy have shown a significant number of new cellular and molecular targets. As a result, a better understanding of these preclinical interpretations can help unravel novel biological and pharmacological targets for macrovascular complications linked to hyperglycemia [30].

### 2.4. Microangiopathy

Diabetic microangiopathy refers to abnormal small vessels found in many organs and tissues in diabetes mellitus. Organs affected by microangiopathy specifically are the eyes, skin, kidneys, and muscles, which are considered the primary factors to determine the prognosis of individuals with diabetes mellitus. Microangiopathy may contribute to neuropathy, retinopathy, and nephropathy that often occur in diabetes [40]. In addition, it may be involved in the pathogenesis of necrobiosis lipoidica, erysipelas-like erythema, and pigmented pretibial patches that occur in patients with diabetes mellitus. Small vessel disease may contribute as much as atherosclerosis of large vessels in causing gangrene to the lower limbs and feet in diabetic patients [40]. In many cases, microangiopathy is clinically silent [40]. Despite the major role of complications of microangiopathy in diabetic patients, the significance of microangiopathy and small vessel damage in the pathogenesis of diabetic foot ulcer is still indefinable and debatable [41]. Nonetheless, different functional and structural microvascular changes occurring in diabetic patients might increase the vulnerability of the skin or can contribute to impaired wound healing [41].

## 3. Treatment Therapies for Diabetic Foot Ulcers

Diabetic foot ulcers typically result from two or more risk factors happening together. Risk factors include neuropathy, PAD, and foot deformity, accompanied together with an external trauma such as poorly fitting footwear or a trifle injury to the foot, can lead to DFU over time [42]. The primary aim of DFU management is wound closure aiming to treat DFU at an early stage to allow rapid healing (Figure 1). The essential components for DFU management are: treating the underlying disease processes, ensuring adequate blood supply, local wound control including infection control, and pressure offloading [43]. Effective foot care should be in partnership between patients, caregivers, and healthcare professionals [43,44,45]. This implies providing the appropriate information to aid patients and caregivers in making decisions, understanding the rationale behind some of the clinical decisions, and supporting good self-care. Treatment of the underlying disease processes starts when the practitioners identify the cause of DFU and try to correct or eliminate it when possible. For instance, treating severe ischemia is essential for wound healing regardless of other interventions [46]. All patients with acute limb ischemia including rest pain, ulceration, and tissue loss should be referred to the consideration of arterial reconstruction [47]. Another strategy for treating the underlying diseases is attaining optimal diabetic control through tight glycemic control and managing other risk factors such as high blood pressure, hyperlipidemia, and smoking. Managing nutritional deficiencies is also important [42]. Moreover, it is important to address the physical cause of the trauma and examine the foot. Examining the foot is by checking the patient’s proper fit for footwear, wear and tear, and for the presence of any foreign bodies such as small stones, drawing pins, glass fragments, and pet hairs that can all contribute to foot trauma [44]. Ensuring adequate blood supply to the foot is another important parameter in DFU management. Acute limb ischemia is a clinical emergency and may be a great risk if not managed at the right time. As a result, critical limb ischemia, decreased perfusion, or impaired circulation is treated by revascularization in order to achieve and maintain healing, and to avoid or delay a future amputation [48]. In addition, the European Wound Management Association (EWMA) states that “the emphasis in wound care for DFUs should be on radical and repeated debridement, frequent inspection and bacterial control and careful moisture balance to prevent maceration” [49]. EWMA has provided a document on wound bed preparation that suggests a specific time framework for management. Management should occur through tissue debridement, inflammation and infection control, moisture balance by optimal dressing selection, and epithelial edge advancement [49].

As for pressure offloading, it is very important to offload at-risk areas of the foot in patients with peripheral neuropathy in order to redistribute pressures evenly [50]. Insufficient offloading leads to tissue damage and ulceration. The optimal strategy for offloading is the total contact cast (TCC). TCC is a foot and lower foot cast that is well-molded, and that distributes pressures evenly over the entire plantar surface of the foot. As it is not easy for the patient to remove it, this ensures compliance [51]. Using this cast in patients with a unilateral uncomplicated plantar ulcer can reduce healing time by about six weeks [52]. However, the TCC can have disadvantages, which include: (i) application, which should only be performed by fully trained and experienced professionals; skin irritation and further ulcers can emerge if applied improperly; (ii) prevention of daily inspection, which can leave signs of spreading infection unnoticed; (iii) possibility of sleep disturbance; (iv) difficulty in having a bath; (v) intolerance by patients especially in warm climates; (vi) limitation of patient’s ability to work, and (vii) high cost and low availability [43]. Another treatment used for DFU is Becaplermin, a recombinant platelet-derived growth factor, which is the only drug approved by the Food and Drug Administration (FDA) to treat diabetic neuropathic ulcers [53]. Its biological activities are similar to the endogenous platelet-derived growth factors. These activities include improving chemotactic recruitment of cells being involved in wound repair in addition to promoting cell proliferation and angiogenesis, and enhancing granulation tissue formation [53,54]. However, Becaplermin has several problems including low systemic bioavailability and promoting malignant tumors far from the application site, in addition to the uncertainty of the validity for diabetic ischemic ulcers. Therefore, it is important to explore new treatments to deal with the effects of DFU. The delivery efficiency of growth factors can be enhanced now via cell therapy [53].

## 4. Stem Cell Therapy

The conventional treatment for diabetic foot ulcer is mainly medical treatment and surgical intervention (endovascular and/or open) for blood flow reconstruction. Nevertheless, medical treatment cannot solve the problem when it comes to foot ischemia caused by arterial stenosis and occlusion. Surgical treatment also faces some problems to restore the blood flow in some cases [53]. Preferred methods for lower limb ischemia management are percutaneous angioplasty and stent therapy [55]. These procedures are difficult to perform in below-the-knee long segment lesions in patients with microvascular pathology. For instance, many patients lack the distal arterial outflow tract [56], and the lower extremity artery lesions in DFU involve the lower leg artery [55]. For this reason, these patients face the risk of amputation, as they are unable to receive interventional therapy for arterial bypass, which is also a risk in specific patients that suffer cardiovascular and cerebrovascular diseases. Innovative and effective therapies have been developed to help repair chronic wounds; these therapies include topical applications of growth factors and cell-based therapies [53]. Although both growth factors and stem cell therapies have a defect in fully repairing DFU, stem cell therapy still has an advantage over growth factors in DFU as stem cells can regulate tissue regeneration in an all-around way by refining the microenvironment at the wound site [53]. It has been reported that stem cells can have an influence on many pathophysiological processes such as ulcer healing by stimulating the tissue repair cell activities, increasing the synthesis of the extracellular matrix, and promoting angiogenesis in the ischemic tissue [57,58,59]. Some animal studies have proven that stem cell implantation results in improved blood flow circulation in ischemic limbs [59,60,61]. Moreover, recent studies have reported that stem cell therapy could reduce amputation rate. However, the association between stem cell therapy and chronic wounds that lead to amputation has not been discussed thoroughly [59,62]. Chiang et al. mentioned that studies with a meta-analysis reporting a relationship between chronic wounds and stem cell therapy were investigating one etiology at a time, such as diabetes mellitus and critical limb ischemia [63,64]. In addition, stem cells play a major role in post-injury and routine homeostasis skin repair [53]. Recently, light has been shed on stem cell therapy as the new technique for treating diabetic lower limb ischemic disease, and specifically, diabetic foot ulcer. Stem cell therapy aims to support wound healing by stimulating the formation of new blood vessels that increase blood supply and relieve limb ischemia. The role of stem cells is better expressed in repairing damage after being administered to traditional treatment; this in turn improves the negative consequences of severe complications of DFU [53]. Stem cells involved in DFU therapy include autologous, allogeneic, and xenotransplantation cells.

### 4.1. Autologous Stem Cells

Autologous stem cell therapy (ASCT) has emerged as a novel promising therapy for ulcer healing, and for those who suffer from lower extremity chronic wounds [57,58,59,63]. The first human trial in 2002 suggested that bone marrow mononuclear cell implantation was effective and safe for therapeutic angiogenesis in patients with critical limb ischemia [59]. This cell implantation was able to help with complete ulcer healing and reduce amputation rate [59,65]. Subsequently, more evidence was coming out indicating that ASCT was more effective than standard therapy for lower extremity chronic wounds [59,66]. Autologous stem cells used in therapy include bone marrow-derived stem cells (BMSCs) such as bone marrow mesenchymal stem cells (BMMSCs) and bone marrow mononuclear cells (BMMNCs), peripheral blood-derived stem cells (PBSCs), peripheral blood mononuclear cells (PBMNCs), adipose-derived stem cells (ASCs) [59], human processed lipoaspirate (PLA), bone marrow-enriched tissue repair cells (BMTRCs) [63], and autologous adipose-derived stromal vascular fraction (SVF) cells [67].

#### 4.1.1. Bone Marrow-Derived Stem Cells

Bone marrow is considered a good candidate for chronic wound treatment as it contains MSCs, inflammatory cell progenitors, and multipotent stem cells. Inflammatory cells are known to take part in wound healing [68,69], and hematopoietic hormones such as granulocyte colony-stimulating, have also been reported to accelerate wound healing [70]. As for MSCs, they can fill the skin dermis, and it has been reported that they can be phenotypically altered or senescent in chronic wounds [71]. Considering the plasticity of bone marrow cells, it is possible that they can produce new skin cells [72]. BMMSCs are the most used cell type in clinical and preclinical studies. Bone marrow cell collection is an invasive and costly method that requires aspiration, density gradient centrifugation, and frequent maintenance and growth of the cells [71]. Two simple methods for BMMSCs isolation, called Ficoll and Harvest system, have been developed to include adequate cell population and similar therapeutic results. Different studies have reported the efficacy of BMMSCs in chronic wound treatment. For instance, topical application of bone marrow stem cells has improved chronic wound healing within 2–4 weeks post use [73]. There was also a positive correlation between the number of cells applied and the percent decrease in wound size, suggesting that bigger wounds need a larger number of MSCs [74]. Badiavas and Falanga reported that directly applied bone marrow-derived stem cells lead to dermal rebuilding and complete closure of chronic wounds in all patients in the study [72]. Another study by Dash et al. reported that treatment with autologous BMMSCs was simple, safe, and effective therapy for chronic non-healing wounds [75]. This therapy was able to control pain, decrease ulcer size, and increase the pain-free walking distance. All biochemical parameters stayed in the normal range post treatment indicating no negative effect of the treatment on the body [75]. Wu et al. reported that the combination of autologous platelet-rich gel (APG) and BMMSCs was able to help in healing DFU and tissue regeneration [76]. In addition, an injection of bone marrow concentrate intramuscularly into the diabetic foot for patients with critical limb ischemia saved 80% of the legs in the study with a significant improvement in toe pressure, tissue perfusion, and better metabolism of critically ill legs [77]. In short, bone marrow-derived MSCs are proving to be a suitable adjunctive therapy to accelerate wound healing and promote limb salvage. In patients with end-stage critical limb ischemia due to peripheral artery disease, bone marrow cell transplantation was safe and was able to improve leg perfusion, significantly reduce major amputations and permit long-lasting limb salvage [78]. Studies have also been conducted to study the effect of BMMSCs on wound healing, and positive effects were reported. For example, Matoba et al. reported that BMMSCs can lead to long-term improvement in limb ischemia with elongation of the amputation-free interval [79]. In a randomized study conducted by Lu et al., the effect of intramuscularly injected BMMNCs and BMMSCs in the treatment of chronic wounds in patients with diabetic critical limb ischemia and foot ulcers was compared [80]. The results showed that both cell types were able to decrease pain and improve perfusion on the ankle-brachial index (ABI) and transcutaneous partial pressure of oxygen (TcPO2). However, the ulcer-healing rate was significantly higher in the BMMSC treated group six weeks post injection, and 100% healing was reached four weeks earlier than in the BMMNC group. In addition, 24 weeks post treatment, more significant improvements in limb perfusion were observed in the group with injected BMMSCs compared to those treated by BMMNCs in terms of painless walking time, ABI, TcPO2, and magnetic resonance angiography (MRA) analysis. These results indicate that BMMSCs can be better tolerated and more effective than BMMNCs for accelerating healing of foot ulcer and in promoting limb perfusion in diabetic patients with critical limb ischemia [80].

#### 4.1.2. Peripheral Blood Stem Cells and Granulocyte Colony-Stimulating Factor

When applied in critical limb ischemia patients with DFU, peripheral blood stem cell treatment has proven to promote healing and prevent amputation. Scantena et al. evaluated PBMNCs’ treatment compared to standard treatment in no-option critical limb ischemia (NO-CLI) with DFU. They reported that significantly lower amputation rates were observed in the group treated with PBMNCs compared to the control group. In addition, results showed that at two years follow-up, 80% of the follow-up group were still alive while only 20% of the control group survived, indicating that PBMNCs reduce amputation, improve survival, and wound healing [81]. Yang et al. conducted a study in which autologous PBSCs were injected as treatment into the ischemic lower extremity and foot, intramuscularly in patients with DFU and lower extremity ischemic disorders. The results showed that autologous PBSC transplantation was able to prevent amputation of the lower extremity of the foot and improve quality of life, and this method might be safe and effective for lower extremity ischemic disorder [82]. To be able to use autologous PBSCs, their production should be increased, which can be achieved by the aid of granulocyte-colony stimulating factor (G-CSF). G-CSF is an endogenous hematopoietic growth factor that induces terminal differentiation and release of neutrophils from the bone marrow [83]. G-CSF promotes the growth of both normal and defective neutrophils and improves their function in patients with diabetes [84,85]. It is also discussed that G-CSF plays a central role in host response to infection [86] and has immunomodulatory and antibiotic-enhancing properties [87,88]. G-CSF is a cytokine able to stimulate the bone marrow and mobilize endothelial progenitor cells (EPC) thereby increasing their number to heal DFU; G-CSF is usually found in wound tissue in acute injury [89,90]. As EPCs circulate in low concentrations in the steady state, G-CSF is a major addition to increase PBSC production obtained for therapeutic use. G-CSF has been used in clinical studies, and it has shown to promote wound healing and reduce surgical interventions in patients with DFU [88,90,91]. G-CSF is usually injected subcutaneously, intramuscularly, or intravenously, and is rarely applied topically [88].

#### 4.1.3. Adipose-Derived Mesenchymal Stem Cells

Due to its subcutaneous location, easy accessibility, abundant sources, and longer incubation time that allows the proliferation ability and differentiation compared to BMMSCs, adipose-derived mesenchymal stem cells (AMSCs) have become a substitute for cell therapy. AMSCs can be autologous, allogeneic, or xenogeneic, and can be harvested with less pain and a less invasive procedure compared to BMMSCs. Furthermore, they have an approximate of threefold increase in immunosuppressive activity [92]. Moreover, AMSCs are considered ideal as a cell-based therapy for chronic conditions due to the fact that they have a stronger cell division capacity than BMMSCs [93,94]. Since AMSCs are harvested from autologous adult fat, fewer ethical controversies exist [95]. In addition, several studies have shown that AMSCs are able to promote wound healing via increasing epithelization and granulation tissue formation, via their anti-inflammatory and anti-apoptotic effects, and through the release of angiogenic cytokines [96,97]. A study by Cianfarani et al. reported that diabetes mellitus impairs the function of AMSCs and alters their intrinsic properties, which affects their healing properties in DFU in diabetic rat models [98]. Moreover, AMSCs from diabetic mice released less vascular endothelial growth factor A (VEGF-A), less hepatocyte growth factor (HGF), and less insulin-like growth factor-1 compared to non-diabetic mice. Kim et al. compared the degree of wound healing for those treated with normal AMSCs and AMSCs derived from diabetic mice. They have reported that dermal regeneration, granulation tissue formation, keratinocyte proliferation, reepithelization, and wound healing rate were higher in the normal AMSCs group. Nevertheless, the diabetic AMSCs were able to retain their ability to stimulate angiogenesis and neovascularization when compared to the control group [99]. On the other hand, Rennert et al. reported that diabetic AMSCs are ineffective in promoting neovascularization and wound healing. This proposed that the use of autologous AMSCs in cell-based therapy in diabetic patients is limited, and interventions are needed and necessary to improve cell function prior to application [100]. Fromer et al. evaluated the ability of human endothelial cell secretome to reverse the damaging effects of high glucose concentrations on AMSCs through priming, which improves their ability to take part in healing and angiogenesis in in vivo murine models, yet this is still to be simulated in clinical trials [101]. Moreover, studies assessing wound healing have reported beneficial effects of adding platelet-rich plasma to AMSCs, specifically acting as a powerful paracrine effector and cell carrier to increase the potential of transplanted cells used in therapies [102]. In addition to diabetes, obesity and aging negatively impact AMSCs, impairing their function and properties including angiogenesis, multipotent state, differentiation, metabolism, and immunomodulation [95,97,103,104,105]. As a result, it can be deduced that employing allogeneic normal AMSCs rather than autologous, diabetic, and impaired ones is a better alternative in cell therapy to promote wound healing in diabetic patients [99]. Furthermore, Khalil et al. assessed the effectiveness of AMSCs embedded in autologous platelet-rich fibrin (PRF) on chronic DFU healing by comparing it to the treatment with PRF alone. They used this protocol as it has been assumed to be a cost-effective therapeutic approach, and better and faster than conventional chronic wound treatments [106]. Khalil et al. reported that patients receiving AMSCs plus PRF had efficient wound healing compared to the patients receiving PRF alone. Their results were consistent with other studies proposing that MSCs secrete high levels of VEGF and HGF, and maintain a higher ratio of transforming growth factor (TGF)-β3 to TGF-β1; this contributes to cell migration, proliferation, improvement of vascular angiogenesis, and matrix deposition [106]. They also promote wound reepithelization, modified collagen deposition, and reduce scar formation [106,107].

### 4.2. Allogeneic Stem Cells

Stem cells isolated from individuals of the same species but not from the recipient are considered allogeneic. Placental, amniotic, embryonic, umbilical cord (UC), and umbilical cord blood are allogeneic sources, from which pluripotent mesenchymal stromal cells are harvested [71]. Placental and amniotic fluid MSCs have a unique potential to differentiate, and they are easy to access which gives them a great clinical value in regenerative therapies. Placental MSCs have been used to treat ischemic complications of chronic wounds while umbilical cord blood mesenchymal stem cells (UCMSCs) have been applied to accelerate cutaneous wound healing. Several animal studies have been conducted using placental mesenchymal stem cells and very few were conducted in humans. The advantage of using placental or umbilical cord MSCs is the ease of obtaining them in a non-invasive procedure [108].

#### 4.2.1. Human Umbilical Cord Mesenchymal Stem Cells

BMMSCs transplantation has been used in therapeutic angiogenesis to treat pathological conditions such as cerebral infarction [109], myocardial infarction [110], and limb ischemia [55,111]. However, harvesting of bone marrow stem cells depends on the overall health of the recipient [55]; as a result, umbilical cord blood or peripheral blood is more commonly used since they are easier to harvest [112]. Human umbilical cord mesenchymal stem cells (HUCMSCs) are primary cells that have multi-lineage potential to differentiate into nerve, muscle, and blood vessels, and can support other interventions. Due to their pluripotency, HUCMSCs are considered high profile. Recently, HUCMSCs have been used for injury repair in different clinical fields [113]. Some studies have shown that HUCMSCs can enhance the formation of new blood vessels and strengthen tissue regeneration [113]. In two different studies, Qin et al. used HUCSCs in the treatment of DFU after angioplasty. They have reported that patients treated with HUCMSCs had better wound healing compared to patients that have only undergone angioplasty. The combination of both treatments was able to improve blood supply, reduce amputation rate, promote ulcer healing, and improve the quality of life for diabetic patients in an advanced stage. This indicates that HUCMSCs’ transplantation after angioplasty is a safe and effective clinical therapy for the treatment of severe diabetic foot ulcer [55]. In addition to this mentioned data, research focusing on the mechanism of HUCMSCs revealed that exosomes derived from them are stable and immunogenic, and are able to transport proteins and growth factors with diverse functions to deploy different effects [114]. Studies have also shown that exosomes derived from HUCMSCs can modulate proliferation and differentiation of BMMSCs [115] in addition to their ability to regulate oxidative stress and hinder cell hypoxia damage [116]. According to these results, exosomes are considered promising for the promotion of angiogenesis in wound healing [113]. Yan et al. investigated the effects of HUCMSCs and HUC-Exos (exosomes secreted by HUCMSCs) on diabetic wound healing. They were able to show that HUCMSCs could regulate the proliferation and functional activity of endothelial cells, and promote injury healing by secreting exosomes. HUC-Exos were able to accelerate wound healing by enhancing angiogenesis and ameliorating oxidative stress. This study proposed that addressing diabetic cutaneous wounds can be achieved through a mixed injection, which could reduce the medical and economic burden. In addition, this therapeutic approach could be promising for treating diabetic wounds in the future [113].

#### 4.2.2. Placental-Derived Mesenchymal Stem Cells

There are not so many studies up to date that have been conducted in humans using placental-derived mesenchymal stem cells (PDMSCs) for the treatment of diabetic foot ulcer. At present, bone marrow stem cells are the main mesenchymal cell source, yet placental stem cells are a better choice for different reasons. Firstly, placental stem cells are easily acquired and raise no ethical issues; secondly, larger amounts can be isolated from the placenta compared to bone marrow, and lastly, human placenta has less immunogenicity [108,117]. PDMSCs have been reported to have a great capacity to differentiate into multiple cell types in addition to their secretory capabilities that can promote wound healing [108,117,118,119]. In a clinical study conducted by Zeng et al., the effect of PDMSC hydrogel on diabetic foot ulcer healing was investigated. Their results showed that within 3 weeks, the application of the PDMSC hydrogels lead to a decrease in wound size, shortening of the wound healing duration, and formation of thick granulation tissue that promotes wound healing [120]. Furthermore, this treatment was also able to avoid lower extremity amputation in the DFU patient. The efficiency of the healing effect of PDMSCs in DFU is associated with secretion of paracrine factors, stimulation of vascular differentiation, and immunomodulation. Zeng et al. also proposed that additional studies are still needed with a larger number of patients [120]. Moreover, Du et al. demonstrated the ability of PDMSCs to improve ulcer healing by the production of different cytokines and HGF, in addition to the paracrine actions that stimulated angiogenesis, which participates in wound healing [121]. A recent study by Meamar et al. has been conducted using nanofibers containing PDMSCs plus platelet-rich plasma (PRP) for healing DFU. In this study, the 12-week ulcer treatment was divided into three groups; the first group received PDMSCs, the second group received PDMSCs after being coated with PRP gel, and the third group was the control that received the standard wound care [122]. When comparing the outcome from the three groups, Meamar et al. reported that there was a 66% and 71% reduction in wound size in the groups treated with PDMSCs and PDMSCs plus PRP gel, respectively, while only 36% of wound reduction was observed in the control group. A significant improvement was observed in the wound closure and pain-free walking distance in the groups treated with PDMSCs and PDMSCs plus PRP gel compared to the control group. In addition, the biopsy results showed development of new capillaries in both groups treated with PDMSCs and PDMSCs plus PRP gel [122]. Another recent study by DaVanzo et al. investigated the outcomes for DFU patients treated with a cryopreserved placental membrane containing viable cells (vCPM) to other Cellular- and Tissue-Based Products (CTPs). The effectiveness of the treatment was measured by the reduction of post-treatment ulcer occurrence, and by the reduction in one-year mortality. The results showed that there was a significant reduction in ulcers for vCMP compared to the treatment with CTP, and the application of vCMP was able to reduce mortality by 2.3 percentage points (13–13.8% change) within one year compared to other CTPs [123]. Additional studies on stem cell therapy for DFU in clinical trials are summarized in Table 1.

#### 4.2.3. Embryonic Stem Cells

Embryonic stem cells (ESCs) are pluripotent, and they are isolated from the inner cell mass of the blastocyst early in development [128] or from the in vitro fertilized embryos [71,129,130] (Figure 2). They are self-renewing cells and can differentiate into any cell line [71]. It has been shown that endothelial cells and endothelial progenitor cells (EPCs) derived from ESCs can secrete growth factors and promote neovascularization and angiogenesis [131,132]. Different studies in rat and mice models have tested the ability of ESCs in diabetic wound healing. For instance, Lee et al. demonstrated that the topical application of undifferentiated ESCs was able to accelerate wound healing in diabetic murine models, and the treated wounds had increased levels of growth factors such as epidermal growth factor (EGF), VEGF, and fibronectin compared to the untreated wounds [133]. In addition, another study by Lee et al. demonstrated that the use of differentiated ESCs topically and subcutaneously was able to improve reepithelialization and accelerate wound healing [133]. Loretelli et al. reported that in vivo topical administration of ESC extract was able to promote wound closure, contraction, and reepithelialization [134]. Moreover, ESC extract led to reduction of inflammatory cells, and increased the repair rate and angiogenesis compared to the controls. These results indicate that ESC extracts are equipped with anti-inflammatory and immunomodulatory properties which aid in wound healing in diabetic mice models [134]. Despite the promising potential for ESCs in wound healing, studies in humans are still limited. The use of ESCs is ethically controversial. Even though ESCs can be obtained without destroying the embryos, it is indeterminate if enough ESCs can result from this method, and whether this will eliminate the ethical issues. Due to the fact that ESCs have a strong proliferative capacity and low differentiation maturity, they may be unsafe since they may cause immune rejection and stimulate tumor formation. As a result, ESCs are infrequently employed in DFU treatment [53,135].

### 4.3. Routes of Administration

There are two different modes of administration for the stem cell therapy in the treatment of DFU. The two routes are local and systemic, and studies have shown that both routes have been effective in the healing of DFU. In general, local injection is the most commonly used method in cell delivery, with intramuscular delivery being the most used in clinical studies, while intradermal and subcutaneous injections are mostly used in preclinical studies [90]. As for topical applications, they have been used more in preclinical practice than in clinical. Topical delivery includes sprays, drops, hydrogels, and scaffolds in which extracellular matrix scaffold has been a subject of interest in research. Extracellular matrix is considered the key modulator for cell maintenance, cell differentiation, proliferation, and self-renewal [90]. Collagen and hydrogel scaffolds can mimic the in vivo environment for stem cells, which increases cell function [136], cell retention, and engraftment [90,137,138,139,140]. On the other hand, systemic cell delivery can be administered endovascularly such as intraarterially or intravenously. Systemic administration can be performed during angioplasty, and it may have an immunomodulatory effect and can optimize glucose homeostasis. However, this administration route can have a high surgical risk, may have poor engraftment, and it is expensive [90]. As for local administration, it is low risk, simple, and in cases other than hydrogels and scaffolds, it is considered inexpensive. However, local cell application may need wound debridement prior to treatment. In the cases of injection and topical spray or drops, there is a risk of high cell death, poor engraftment, and no cell density and spacing control in addition to the risk of infection in the case of intramuscular, intradermal, or subcutaneous injection. In the case of hydrogels or scaffolds, the protocol is of high complexity, but it leads to better retention and engraftment, and better control of cell density and cellular spacing [90]. In a study by Chiang et al., they have compared the efficacy of the therapeutic treatment using autologous stem cells in different administration routes, which were intramuscular, topical, and intraarterial [59]. Their results showed that the group that had the stem cell therapy administered intramuscularly had a significantly higher complete wound healing compared to the group with intraarterial route. Chiang et al. explained that this result could be due to the fact that patients with diabetes mellitus have microvascular complications or arterial occlusion of critical limb ischemia leading to peripheral perfusion. This issue could be avoided in intramuscular administration as cells are transported closer to the wound site [59]. In addition, muscle tissues can support injected cells with oxygen and nutrients, which helps in their survival and improves their function [141]. Studies have also reported that intravascular administration of stem cells could trap them in the lungs, leading to pulmonary embolism. As a result, intramuscular administration seems to be a more safe and effective option for stem cell administration [59]. As for topical administration of autologous stem cells, Chiang et al. reported that it was effective in healing wounds as shown in previous studies, and they stated that topical administration contributes to cell metabolism, differentiation, and migration [59]. Topical administration can also shorten treatment time and improve survival rate of the transplant by stimulating extracellular matrix secretion and tissue regeneration [142,143]. These results suggest that topical and intramuscular administration are more effective than intraarterial [59].

## 5. Cell Secretome: A Promising Therapeutic Alternative in Wound Healing

Cell-based therapies have been studied for decades, and they include skin substitutes (e.g., living fibroblasts and keratinocytes) that have shown promising results in promoting faster wound closure, improved reepithelialization, and superior vascularity and wound height [144]. Nevertheless, these therapies have been reported to be expensive, to have potential tumorigenic, infection, and rejection risks, and to have specific storage conditions in addition to the difficulty in using them within a community [144,145]. Recombinant growth factors have been suggested to be used in impaired healing since non-healing wounds lack cytokines or growth factors. Although they have positive effects on wound healing, their clinical administration was limited as high amounts of growth factors were needed for treatments, and they were expensive to manufacture [144]. Moreover, studies have reported that the main therapeutic benefits of MSCs are not only limited to their cell-to-cell interactions, but to a broad series of bioactive molecules that include exosomes, proteins, nucleic acids, cytokines, growth factors, chemokines, proteasomes, membrane vesicles, and microRNA. These bioactive materials are produced in response to the surrounding environment, and they are known as secretome [132,144,146]. Consequently, the MSC secretome affects the adjacent cells and regulates several biological processes [147]. The primary therapeutic mechanism for MSCs is embedded in their paracrine and trophic properties [144]. Despite that MSCs derived from different organs share similar phenotypic and regenerative characteristics, their secretome differs depending on their origin thus leading to different therapeutic potentials [144,148]. Different studies have also compared the effect of stem cell secretome to fibroblast secretome in wound healing, and have found stem cell secretome to be more effective in increased wound healing compared to fibroblast secretome [149,150]. Due to these different properties, MSC secretome is considered a superior alternative to the expensive cytokine and growth factors’ therapy that only delivers limited amounts of protein to the wounds. Furthermore, the transfer of live cells to cutaneous wounds has different challenges. For instance, the cell viability is decreased to 1–32% when injected through a syringe or needle which can also cause irreversible or fatal damage to the cell membrane [151,152]. This negative effect reverses the potential benefit of cell therapy and introduces a population of necrotic cells that may stimulate an immune response, which could be damaging for the healing process. As such, MSC secretome eludes these difficulties with live-cell administration in stem cells, and it also has the advantage of easier mass production, easier packaging, and transportation. These factors have led the MSC secretome to have a promising therapeutic potential in the treatment of tissue regeneration in wounds [132,144,153].

## 6. Bionanomaterials: A Modality for Stem Cell-Based Therapy Application

Wound healing is a series of different dynamic and complex phenomena. Numerous studies have been conducted depending on the type and severity of wounds. Nanomaterials have shown to be promising in faster wound healing among all other wound healing materials. Nanomaterials have unique physicochemical and biological properties. Some of them can be applied directly to the wound, while others can be incorporated into scaffolds to create hydrogel matrices or nanocomposites that promote wound healing through their antimicrobial and antioxidant properties as well as their selective anti- and pro-inflammatory and pro-angiogenic properties. Due to their large surface area to volume ratio, nanomaterials are not only used as vectors for drug delivery. They have also been used for cell therapy with stem cells to promote wound healing. Nanomaterials have particularly interesting properties that can be exploited in regenerative medicine. By using innovative nanodevices in appropriate formulations, topically applied drugs can better pass through skin layers that are otherwise difficult to penetrate, modulating drug release and improving their solubility [154]. Transport through the skin occurs by three routes, depending on the chemical and physical properties of the drug molecules: trans-appendageal, trans-epidermal, or trans-cellular [155]. However, large drug molecules are unable to penetrate intact skin by passive diffusion along a concentration gradient via intracellular and extracellular pathways. This problem is less pronounced in damaged skin tissue in wounds. Active ingredients can be better delivered to the desired site of action with the help of nanomaterials, thereby promoting cell proliferation and differentiation for improved wound healing. Thus, when therapeutic nanomaterials are used for wound healing, the proliferation of skin stem cells (SSCs) into keratinocytes and fibroblasts can be promoted [156]. Fibroblast gene expression can also be modulated such that adapted extracellular matrix (ECM) production is achieved. Overall, this contributes to improved and accelerated regeneration of the damaged tissue.

Wound healing of diabetic foot ulcer (DFU) represents an important application. Integration of silver or magnesium metal ions has also resulted in improved antimicrobial effects and promotion of angiogenesis in vitro [157]. In in vitro experiments, Daňková et al. were able to improve the proliferation of mesenchymal stem cells (MSCs) by combining polycaprolactone (PCL) nanofibers with integrated magnetic nanoparticles [158]. Another study showed that PCL nanofibers in combination with natural plant extracts from *Myrtus communis* exhibited a protective effect on skin mesenchymal stem cells (SMSCs) aged by UV damage. This fact could be quite significant for skin rejuvenation possibilities [154].

Another bionanomaterial with interesting properties for skin regeneration was developed by Akolpoğlu Başaran et al. [159]: heparin was encapsulated in poly(lactic-co-glycolic acid) (PLGA) nanoparticles and incorporated into sericin/gelatin nanofibers. This system exhibited a high retention capacity of water and a low degradation rate in addition to controlled drug release. This, in combination with the two biopolymers, sericin and gelatin, resulted in an effective drug delivery system for topical skin regeneration applications.

However, nanofibers have not currently been used for a therapeutic application for wound healing in humans. Full regeneration of structural and functional properties of the skin has not been possible to date. Nevertheless, since they represent a promising therapeutic approach, the exploration, as well as improvement of nanofibers for tissue engineering, remains an important challenge [160].

## 7. Conclusions

Research findings suggest that stem cell-based therapy in DFU is promising in the field of regenerative medicine. Their therapeutic mechanisms include promoting angiogenesis, secreting paracrine factors, stimulating vascular differentiation, suppressing inflammation, improving collagen deposition, and immunomodulation. These results prove that stem cell-based therapy is effective in wound healing and decreases amputation rates. Nevertheless, their biological properties are still not very clear, and their safety is still controversial in certain cases. In addition, angioplasty application before stem cell therapy has been shown to be very promising in patients needing revascularization, but it is still not clear if stem cell therapy will always be used as a complement to angioplasty. Scaffolds have also been described to have a positive impact on the healing process of DFU by stem cell therapy. In addition, nanofibers represent a promising therapeutic approach in wound healing. Studies have also reported that different functions and properties of stem cells including angiogenesis, multipotent state, differentiation, metabolism, and immunomodulation are impaired in patients with diabetes. Therefore, employing allogeneic rather than autologous, diabetic, and impaired stem cells could be a better alternative in cell therapy to promote DFU healing. As for the administration route of stem cell therapy in DFU, it has been reported that intramuscular and topical administration are safer and more effective than intravascular administration. Moreover, stem cell secretome has proven to be favorable in healing DFU in different situations. Despite these results, more studies with larger scales and different administration routes are needed as most of the studies reported show results from stem cells applied locally rather than systemically. In short, more studies are required to determine the ideal type of stem cells needed in therapy, their safety, the proper dosing, and the optimal administration route.

## Figures and Tables

**Figure 1 biomedicines-10-01507-f001:**
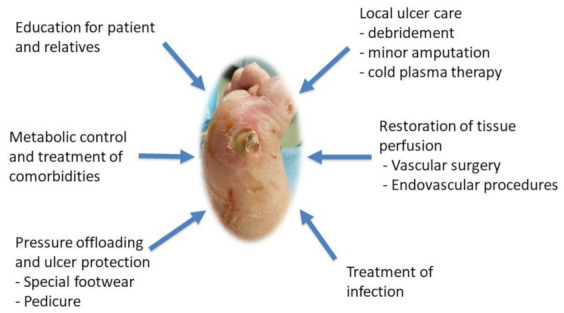
Conventional DFU treatments.

**Figure 2 biomedicines-10-01507-f002:**
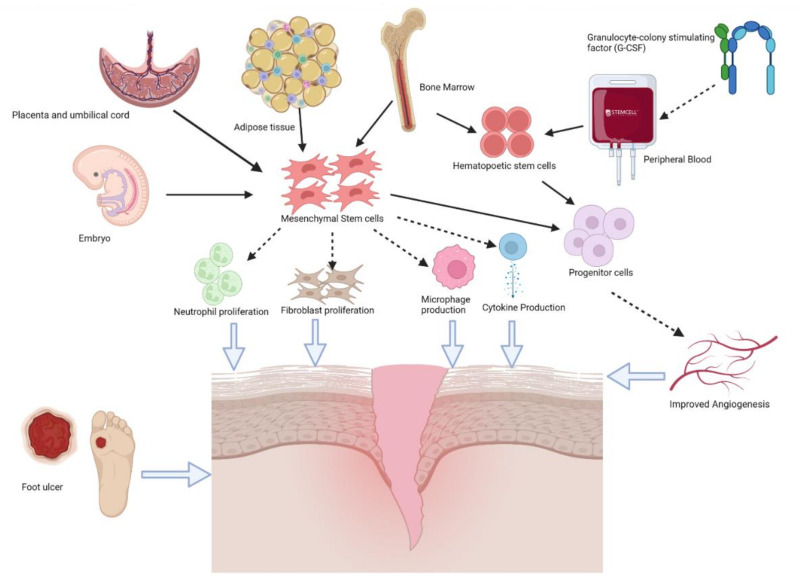
Differentiation of stem cells and their therapeutic potential in DFU healing (created with BioRender). Stem cells differentiate from different sources (e.g., embryo, placenta and umbilical cord, adipose tissue, bone marrow, peripheral blood). Therapeutic mechanism of stem cells in DFU healing includes neutrophil proliferation, fibroblast proliferation, microphage production, cytokine production, and improved angiogenesis.

**Table 1 biomedicines-10-01507-t001:** Stem cell therapy for DFU in clinical trials.

Reference	Study Design	Cell Type	Administration Route	Outcome	Follow-Up
Kirana et al. 2012 [21]	24 patients with DFU 2 treatment groups: - Bone marrow mononuclear cells (BMCs) - Tissue repair cells	Autologous BMMSC	- Intramuscular injection - Intraarterial	- Both groups had improvement in wound healing without a significant difference. - Improvement in TcPO2 was detected in both groups.	45 weeks
Marino et al. 2013 [124]	20 patients with PAD with chronic ulcers of the lower limb - 10 patients treated with AMSCs extracted by celution method	Autologous AMSCs	Local perilesional injection	- Six out of ten had complete healing. - Closure of the ulcer was observed.	90 days
Qin et al. 2016 [55]	53 patients - 2 groups: control group, experimental group - Both groups received angioplasty; those in the experimental group also received HUCMSCs	Allogeneic HUCMSCs	- Intraarterial infusion - Intramuscular	Experimental group had significant improvement in: - Skin temperature - Ankle-brachial pressure index - Transcutaneous oxygen tension- Claudication distance	1–3 months
Xu and Liang 2016 [125]	127 patients were treated with - Granulocyte colony-stimulating factor (G-CSF). - Extracted PBSC suspension	- G-CSF - Autologous PBSCs	Injection into the ischemic lower extremities at multiple points around the embolized blood vessels	Ischemic area of the patients was improved significantly.	4 weeks
Zeng et al. 2017 [120]	57-year old patient with DFU - PDMSC hydrogel	Allogeneic PDMSCs	Topical	- Healing of foot ulcer was observed. - Walking foot function was well preserved.	6 months
Wu et al. 2018 [76]	A 54-year-old patient with DFU. Received standard treatment including debridement, dressing, and continuous negative pressure suction followed by autologous platelet-rich gel (APG) and BMMSC transplantation	Autologous BMMSCs	Local perilesional injection	Significant improvement of wound and complete healing was detected.	30 days
Zhao et al. 2020 [126]	12 patients with DFU	- Allogeneic umbilical cord mesenchymal stem cells (UCMSCs) - Umbilical cord blood-derived endothelial colony-forming cells (ECFCs)	Local injection	- Accelerated healing in wounds treated with combination therapy was observed. - Wound size reduction was detected.	1–4 weeks
Scatena et al. 2021 [81]	76 no-option critical ischemia (NO-CLI) patients DFUs - All patients treated with the same standard care (control group); - 38 patients were also treated with autologous PBMNC implants	Autologous PBMNC	- Intramuscular - Local peri -lesional injection	- Four out 38 amputations (10.5%) in the PBMNC group were done. - 15 out of 38 amputations (39.5%) in the control group (*p* = 0.0037) were done. - At 2 years follow-up, 80% of the PBMNC group was still alive vs. only 20% of the control group (*p* = 0.000). - 33 patients healed (86.6%) in the PBMNC group. - One patient healed in the control group.	2 years
Carstens et al. 2021 [67]	63 patients with type 2 diabetes with chronic DFU—all amputation candidates - Treated with 30 × 106 SVF cells	Autologous adipose-derived stromal vascular fraction (SVF)	- Paravascular injection (pedal arteries) - Local perilesional injection	- At 6 months, 59 of the 63 subjects enrolled were evaluable for closure. - Fifty of the evaluable subjects achieved closure at 12 months (93%; confidence interval = 0.813–0.976). - The remaining four evaluable subjects had wound closure of ≥85%.	6–12 months
Chiang et al. 2021 [59]	Meta-analysis: - Authors assessed randomized controlled trials (RCTs), and extracted data on complete healing rate, amputation rate, and outcomes regarding peripheral circulation. - Extracted data pooled using a random-effects model - A total of 28 RCTs were eligible	Autologous stem cell therapy (ASCT)	Intramuscular	- ASCT significantly improved complete wound healing rate as compared with standard treatment for lower extremity chronic wounds (LECWs). - ASCT could promote the healing of LECWs.	1–5 months
Khalil et al. 2021 [106]	10 patients with an open DFU wound 2 groups: - Group A: injected with PRF alone - Group B: injected with AMSC embedded in PRF	Autologous AMSCs	Topical	Group B had better healing index than group A.	2–4 months
Uzun et al. 2021 [127]	20 patients with DFU 2 groups: AMSCs, standard wound care	Allogeneic adipose-derived mesenchymal stem cells	Local intra -lesional injection	Treatment group had better and faster wound healing compared to control.	48 months

## Data Availability

Not applicable.

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
