# Peer review of "Stem Cell-Based Therapy: A Promising Treatment for Diabetic Foot Ulcer"

_biomedicines, 2022, doi:10.3390/biomedicines10071507_

Round 1

Reviewer 1 Report

In this work, the authors reviewed the different types and origins of stem cells and routes of administration used for treatment of DFU in clinical and preclinical studies. Angioplasty could be complementary to stem cell therapy, and scaffolds have a positive impact on the healing process of DFU by stem cellbased therapy. Stem cell-based therapy is promising in the field of regenerative medicine, but more studies are still needed to determine the ideal type of stem cells required in therapy, their safety, proper dosing, and optimal administration route.

Major Comments:

1. Compared with other reviews that have been published in this field, does this review have any novel expressions?

2. Has the citations of the literature been fully investigated?

Minor comment:

1. The English should be improved, and some typos need to be revised.

2. Mark the full name for the first occurrence of any abbreviation.

Reviewer 2 Report

The Manuscript entitled “Stem cell-based therapy: a promising treatment for diabetic foot ulcer” is a very interesting review of the main types of steam cells that can be used in the treatment or as an adjuvant to the main treatment of the diabetic foot ulcer. The manuscript is well written, scientifically sound, easy to read and certainly will be of interest of the Biomedicines’ public. In this review opinion, the authors should dedicate a short sentence/paragraph to the main microorganism involved in the diabetic foot ulcer infection. The figure and the table are well organized and I couldn’t detect any flaw on them. After the inclusion of this suggestion, the manuscript will be eligible to be published in Biomedicines’ Journal.

Reviewer 3 Report

1. Authors must summarize the table for conventional treatment.

2. schematic representation of treatment from stem cells must be added. 

3. What are the drawbacks of conventional treatment of diabetic foot ulcer.

4. Resolution of Figure is poor.

Round 2

Reviewer 3 Report

Accept